# CONTACT-GUIDED REAL2SIM FROM MONOCULAR VIDEO WITH PLANAR SCENE PRIMITIVES

**Zihan Wang**[*]  **Jiashun Wang**[*]  **Jeff Tan**  **Yiwen Zhao**

**Jessica Hodgins**  **Shubham Tulsiani**  **Deva Ramanan**

Carnegie Mellon University

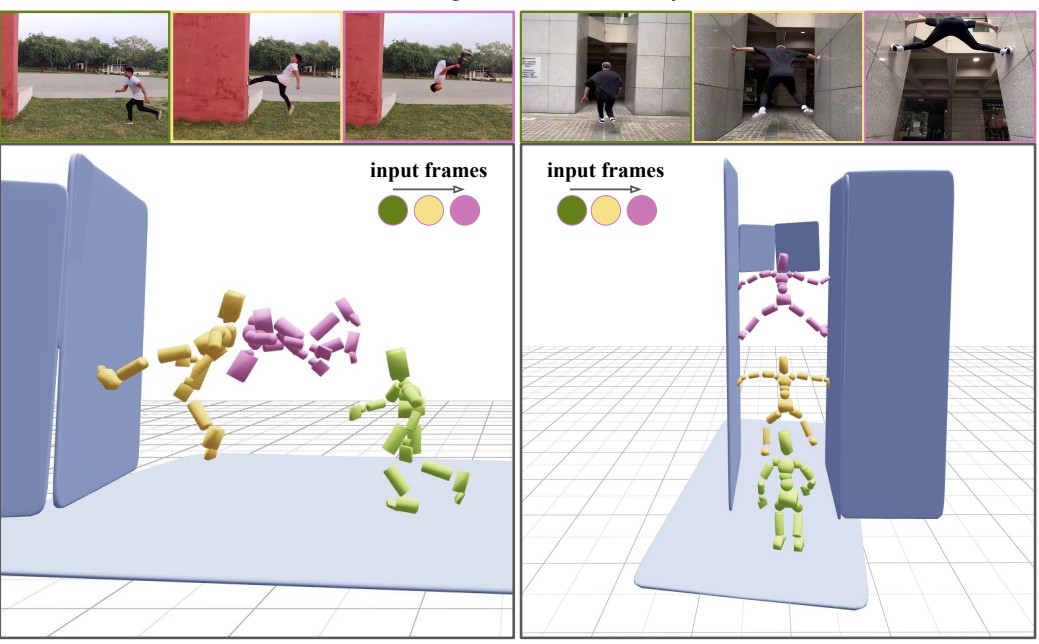

Figure 1: We present CRISP, a method for recovering 3D human motion and *simulatable* scene geometry from monocular video. Our key insight is to fit compact planar primitives (blue) to point-cloud reconstructions of the scene. We then use the joint human and scene reconstruction to train a humanoid controller with RL in simulation (dynamics rollouts of the humanoid are colored as green → yellow → pink for different timesteps). Extensive experiments on EMDB and PROX show that CRISP lowers motion-tracking failure rates by $8\times$ compared to prior art.

## ABSTRACT

We introduce CRISP, a method that recovers simulatable human motion and scene geometry from monocular video. Prior work on joint human-scene reconstruction relies on data-driven priors and joint optimization with no physics in the loop, or recovers noisy geometry with artifacts that cause motion tracking policies with scene interactions to fail. In contrast, our key insight is to recover convex, clean, and simulation-ready geometry by fitting planar primitives to a point cloud reconstruction of the scene, via a simple clustering pipeline over depth, normals, and flow. To reconstruct scene geometry that might be occluded during interactions, we make use of human-scene contact modeling (e.g., we use human posture to reconstruct the occluded seat of a chair). Finally, we ensure that human and scene reconstructions are physically-plausible by using them to drive a humanoid controller via reinforcement learning. Our approach reduces motion tracking failure rates from 55.2% to 6.9% on human-centric video benchmarks (EMDB, PROX), while delivering a 43% faster RL simulation throughput. We further validate it on in-the-wild videos including casually-captured videos, Internet videos, and even Sora-generated videos. This demonstrates CRISP's ability to generate physically-valid human motion and interaction environments at scale, greatly advancing real-to-sim applications for robotics and AR/VR. Code and interactive demos are available at our project website: crisp-real2sim.github.io/CRISP-Real2Sim.

---

[*]Equal contribution.

# 1 INTRODUCTION

What does it mean to understand a video of a human? Although there has been tremendous progress in well-studied tasks such as space-time reconstruction or activity recognition, we argue that true human understanding is physical: a person's foot is not simply stepping down, but is placed in contact with a ground surface to provide support. Indeed, humans constantly have close interactions with their surrounding environments - they sit on chairs, lie on sofas, and climb stairs. Given a casual monocular video of such an interaction, our goal is to build a "vid2sim" pipeline for generating human-scene reconstructions. Concretely, we wish to reconstruct the scene with enough fidelity to accurately *simulate* the human, environment, and their interactions while obeying the laws of physics (e.g., avoiding inter-penetrations, foot sliding, and floating geometry). Doing so would unlock scalable learning for applications such as embodied AI and robotics (Allshire et al., 2025), physically-plausible character animation (Yuan et al., 2023), and AR/VR (Luo et al., 2024).

While there has been tremendous progress in reconstructing scenes (Wang et al., 2025a) and humans (Shin et al., 2024; Wang et al., 2024a; Shen et al., 2024), the interaction between the two is less well-studied. While notable exceptions exist (Liu et al., 2025), prior art still struggles on videos with parallax and occlusions (due to the human body), producing reconstructions with duplicate structures or missing regions (Wang et al., 2025d). Moreover, we find that the accuracy of reconstructions needs to be even more precise for physical simulation; even small amounts of noise in ground plane reconstructions can (literally!) trip up a physical humanoid simulation (see Fig. 4). As such, prior work on driving humanoid simulators with video input focuses on simplistic environments where there is limited interaction with scene geometry (Luo et al., 2023a). A final subtle point is the *efficiency* of simulation; collision detection often requires geometry that is well approximated by convex primitives, which can quickly become expensive for complicated scene geometries.

**Contributions**. We introduce CRISP, a real-to-sim pipeline that converts monocular human videos into simulation-ready assets by integrating human mesh recovery (HMR), 4D reconstruction, and contact prediction. Unlike most reconstruction works, which yield noisy and non-watertight 2.5D geometry, CRISP explicitly builds geometric outputs ready for physics simulation, by fitting ***clean, convex and watertight*** planar primitives to point cloud reconstructions via a clustering algorithm. While conceptually simple, this produces simulation-ready geometry that significantly improves the fidelity of physical simulation. Second, we make use of state-of-the-art monocular depth priors to reduce artifacts such as duplicate structures that plague concurrent methods (Allshire et al., 2025). Third, we improve scene reconstruction by reasoning about occluded geometry with inferred human body shape (e.g., the seated pose can infer an occluded chair seat). To do so, we use vision–language models to detect common human scene interactions such as sitting-on-a-chair. Finally, to produce a physically-valid reconstruction, we use reinforcement learning (RL) to drive a simulated humanoid to follow reconstructed human motions while interacting with the reconstructed scene in simulation.

**Results**. We evaluate CRISP across standard human benchmarks (EMDB, PROX) and demonstrate strong gains in both reconstruction fidelity and real-to-sim performance. CRISP achieves a 93.1% real-to-sim success rate, significantly surpassing baselines that collapse under noisy scene geometry. When integrated into RL training pipelines, CRISP supports a 43% faster simulation throughput compared to dense-mesh approaches, while maintaining physically plausible interactions. Somewhat surprisingly, physical reasoning *improves* the quality of both the human and scene reconstruction. These results highlight that CRISP not only bridges the video-to-simulation gap, but also makes RL training from in-the-wild videos practical and efficient for embodied AI and robotics.

# 2 RELATED WORK

## 2.1 MONOCULAR HUMAN MOTION ESTIMATION.

3D human motion recovery is most widely formulated as recovering the parameters of a parametric human model, such as SMPL (Loper et al., 2023) or SMPL-X (Pavlakos et al., 2019). Classic methods (Bogo et al., 2016) rely on optimization, fitting body model parameters to match the human shape of the input. More recently, feed-forward HMR methods (Kocabas et al., 2020; Shen et al., 2024) directly regress SMPL parameters using deep neural networks such as transformers. Although these works typically recover 3D humans in camera coordinates, recent focus has shifted to recover metric human trajectories jointly with scene geometry and contacts in world frame (Ye et al., 2023; Wang et al., 2024a). For example, TRAM explicitly estimates camera parameters using DROID-SLAM (Teed & Deng, 2022) by masking out dynamic regions, then uses the camera to un-

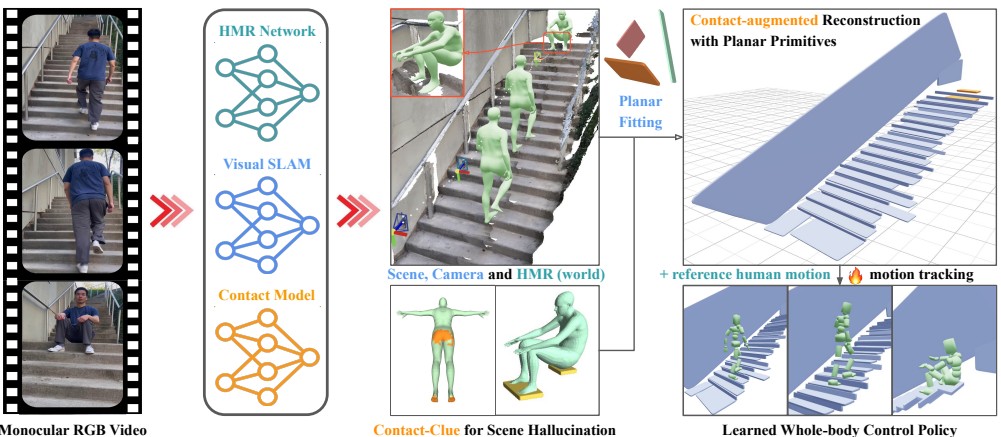

Figure 2: **CRISP pipeline.** Given a casual RGB video (**left**), CRISP reconstructs scene geometry and human motion that are used to drive a humanoid controller in simulation. After recovering camera poses, intrinsics, and global point clouds (**middle-top**), we propose a clustering algorithm to obtain a small number ($\approx 50$) of compact planar primitives that enable efficient simulation (**right-top**). We also detect human-scene contacts and explicitly use them to recover interaction surfaces that may be occluded, such as stair and its platform (**middle-bottom**). Finally, we use our human and scene reconstructions to drive a humanoid controller in simulation via RL (**right-bottom**).

project people into world coordinates. WHAM (Shin et al., 2024) is trained to predict the likelihood of foot-ground contact using estimated contact labels, using these contact estimates to refine the body pose. (Liu et al., 2025) jointly recovers scene geometry, human motion, and contact points, then jointly optimizes all of these observations using human-scene contact constraints. Despite the recent progress in human-scene reconstruction, existing works largely rely on integrating multiple data-driven priors, such as feed-forward HMR and geometric foundation models. Beyond that, we propose to leverage physics simulators to infer simulatable human-scene interactions.

## 2.2 HUMAN-SCENE INTERACTION

Realistic human–scene interaction has long been a central challenge in the computer vision community. To achieve accurate human–scene motion modeling, modeling contact is essential (Wang et al., 2021; Zhang et al., 2020b; Nam et al., 2024). A line of work explicitly works on contact prediction (Huang et al., 2022; Dwivedi et al., 2025). On the other hand, physics-based control methods have gained significant attention for their ability to produce natural and physically plausible interactions, thanks to physics simulations. For example, Chao et al. (2021) constructed a library of policies capable of tasks such as chair sitting by imitating motion capture trajectories. Yu et al. (2021) targeted more dynamic behaviors, training distinct controllers to reproduce complex parkour movements captured from video. Beyond direct motion tracking, adversarial imitation frameworks encourage physically grounded behaviors in indoor environments, leading to more natural and diverse interactions with objects and surfaces (Hassan et al., 2023; Xiao et al., 2023). Luo et al. (2022) introduce embodied scene-aware human pose estimation, where an agent equipped with proprioception and scene awareness recovers human motion in simulation; however, the surrounding scene is given rather than reconstructed. HIL (Wang et al., 2025b) explores hybrid imitation learning by combining motion tracking with adversarial learning to train unified parkour controllers from Internet videos, but requires manual scene annotations to function effectively. Most relevant to our work is the concurrent VideoMimic framework (Allshire et al., 2025), which proposes a real-to-sim-to-real pipeline that jointly reconstructs humans and environments, producing control policies for humanoids capable of skills such as sitting and walking. Compared with VideoMimic, our approach provides more accurate modeling of humans, scenes, and contacts, resulting in substantially improved simulation stability, efficiency, and RL success rates.

## 3 METHOD

### 3.1 OVERVIEW

Given a casually-captured monocular video $\mathcal{V} = \{ I_i \in \mathbb{R}^{H \times W} \}_{i=1}^{N}$ depicting a human interacting with a *static* scene $\mathcal{S}$ (e.g. parkour, stair climbing, or sitting on a sofa), our goal is to recover

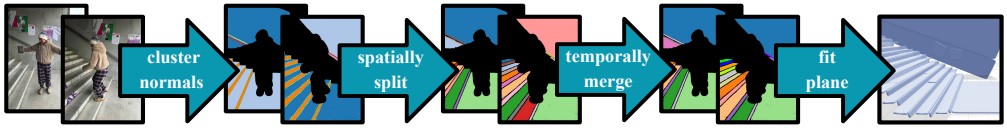

Figure 3: **Planar fitting.** Given per-frame pointmaps from a visual SLAM system, we (1) produce candidate planar segments by running K-means on *normal maps* (computed via finite-differences on pointmaps); (2) *spatially-split* segments via DBSCAN on 3D points within each segment; (3) *temporally-merge* segments across frames with similar planar fits and sufficient optical flow correspondences (Zhang et al., 2025). Notably, physical planes that may appear as multiple segments in different frames are merged into a single temporally consistent planar region (pink and blue segments before merging → blue after merging). We then fit a plane to each merged planar region using RANSAC and define a planar cuboid with a default thickness of 0.05m. See Appendix B for details.

*simulatable* 3D human motion and scene geometry in a common world coordinate. We begin by inferring camera poses, intrinsics, and a global scene point cloud. Then, to obtain *simulatable* compact planar primitives from the global scene point cloud, we propose a simple clustering algorithm (Sec. 3.2). We make use of human-scene contact modeling (Sec. 3.3) to recover scene geometry that might be occluded during interactions. Finally, we ensure that human and scene reconstructions are physically plausible by using them to drive a humanoid controller via RL (Sec. 3.4).

**Human, Scene, and Camera Initialisation** Similar to concurrent work (Allshire et al., 2025), we use MegaSAM (Li et al., 2024) to jointly recover camera intrinsics $\mathcal{K} \in \mathbb{R}^{3\times3}$, per-frame camera poses $\mathcal{T}_i = [\mathcal{R}_i \mid t_i] \in SE(3)$, and a per-frame dense depth map $\mathcal{D} = \{(d_i)\}_{i=1}^N$ from unconstrained monocular video. To improve the geometry quality, we replace the depth estimator in the optimization stage of MegaSAM with MoGe (Wang et al., 2025c), producing a *scale-invariant* dense point cloud $\mathcal{P}$ together with calibrated camera parameters $\{\mathcal{K}, \mathcal{T}_i\}_{i=1}^N$. To estimate 3D human pose, we pass the intrinsics $\mathcal{K}$ to GVHMR (Shen et al., 2024) to obtain an SMPL mesh in camera space, then lift the 3D humans to world frame using the estimated camera poses $\mathcal{T}_i$, ensuring that the human, scene, and camera share a single coordinate system. Although MegaSAM reconstructs $\mathcal{P}$ up to an unknown scale factor, the scale of a typical human is known: we use this cue to recover a metric-scale point cloud $\tilde{\mathcal{P}}$ by scaling $\mathcal{P}$ such that depth of the human in the scaled MegaSAM point cloud matches the depth of the 3D SMPL mesh from GVHMR.

### 3.2 NORMAL–BASED PLANAR PRIMITIVE FITTING

**Why planar primitives?** Although reconstruction pipelines typically output point clouds, physics-based simulators (e.g. Isaac Gym) require meshes for collision detection and force calculation. Typical pipelines convert point clouds to 3D meshes by fusing points into a truncated signed distance function (TSDF), then meshing via Marching Cubes. However, not only are these meshes very large (with hundreds of thousands of triangles), but these pipelines often produce *noisy* meshes, with oversmoothed surfaces in some regions and unwanted geometry artifacts in others. Such artifacts can cause serious issues when driving a humanoid controller in simulation (Fig. 4). The humanoid may bump into reconstruction artifacts and experience unstable contact forces, causing RL-driven humanoids to fail to reproduce the motions observed in video.

**Our key insight** is that decomposing the scene into a small set ($\approx 50$) of convex primitives can solve both issues. Convex primitives are small and efficient to simulate, and standardized primitives also regularize the reconstruction, making it more robust to low-level noise. Our specific choice of primitive is based on a *planar-world assumption*: many human-scene interactions such as sitting, lying down, parkour, climbing stairs, etc. can be represented as interactions with planar surfaces. While prior work has investigated planar decompositions via a combination of 2D segmentation priors and global neural fields (Ye et al., 2025), we find it sufficient to simply cluster 3D point cloud reconstructions (from visual SLAM) into planar primitives (Fig. 3), yielding an efficient, lightweight, and simulation-ready reconstruction without per-scene optimization.

### 3.3 CONTACT AS A CUE FOR SCENE COMPLETION

**Why contact?** Given an unconstrained monocular video, critical interaction surfaces in the scene might be occluded by the human or other scene geometry. For example, a person might stand on a ground plane that is out of view, or sit onto a couch that is now occluded by their body. From

an input frame $I_t$ where the posed SMPL mesh is estimated as $\mathcal{M}_t$, we aim to estimate per-vertex contact predictions $c_t(v) \in \{0, 1\}$ and use them to guide the completion of scene geometry. Given an image of a person (potentially interacting with their environment), InteractVLM (Dwivedi et al., 2025) predicts a binary contact mask over SMPL vertices that are in contact with the scene. When naively applying InteractVLM to frames of a video, we tend to find it over-predicts false positives during "near-contact" frames, presumably because it was not trained on such hard negatives.

**Temporal–kinematic filtering.** To reduce false positives, we apply non-maximum suppression to contact predictions across time. Specifically, we keep only those predictions with consistently high confidence for $L$ frames and return the frame $t$ with the smallest amount of human motion $v_t$:

$$t^* = \underset{t \in \{i, i+L\}}{\arg \min} \ v_t$$

## 3.4 PHYSICS-BASED MOTION TRACKING

Following Peng et al. (2018), we train a fully-constrained motion-tracking policy $\pi^{\text{FC}}$ to imitate the full-body motion sequence extracted by our pipeline. Specifically, the policy takes as input the character state $s_t$ and the next $K$ target poses $g_t = [f_t, ..., f_{t+k}]$. The output is an action $a_t$ to let the character track the reference motion precisely. The policy is trained using a standard motion tracking reward $r$, which encourages the character to minimize the difference between the state of the simulated humanoid and the reference motion at each timestep $t$. We follow MaskedMimic (Tessler et al., 2024; Wang et al., 2025b) in the design of observation, action, and reward of the model.

**Observation** The simulated character is constructed based on the SMPL human model (Loper et al., 2023; Luo et al., 2023b). The robot state is represented by a set of features that describes the configuration of the character's body,

$$s_t = \left( \theta_t \ominus \theta_t^{\text{root}}, \ (p_t - p_t^{\text{root}}) \ominus \theta_t^{\text{root}}, \ v_t \ominus \theta_t^{\text{root}} \right),$$

where $\theta_t$ and $p_t$ are joint orientations (quaternions) and positions, $v_t$ are the linear and angular velocities, and $\ominus$ represents quaternion subtraction. The policy is additionally conditioned on the next $K$ target poses $g_t = \left[ \hat{f}_{t+1}, \hat{f}_{t+2}, \ldots, \hat{f}_{t+K} \right]$, with joint-wise targets $\hat{f}_t^j = \left( \hat{\theta}_t^j \ominus \theta_t^j, \ \hat{\theta}_t^j \ominus \theta_t^{\text{root}}, \ (\hat{p}_t^j - p_t^j) \ominus \theta_t^{\text{root}}, \ (\hat{p}_t^j - p_t^{\text{root}}) \ominus \theta_t^{\text{root}} \right)$.

**Action.** Following prior work (Peng et al., 2018; Tessler et al., 2024), actions are parameterized as desired joint targets for a Proportional–Derivative (PD) controller. The stochastic policy $\pi\left(a_t \mid s_t, g_t\right)$ is modelled as a multivariate Gaussian with fixed diagonal covariance matrix $\Sigma_\pi \ \sigma_\pi = 0.055$.

**Reward.** The reward function is defined as

$$r_t = w_p e^{-\alpha_p ||\hat{p}_t - p_t||} + w_r e^{-\alpha_r ||\hat{q}_t \ominus q_t||} + w_v e^{-\alpha_v ||\hat{\dot{p}}_t - \dot{p}_t||}$$
$$+ w_\omega e^{-\alpha_\omega ||\hat{\dot{q}}_t - \dot{q}_t||} + w_h e^{-\alpha_h ||\hat{h}_t - h_t||} + w_e \sum_j ||\tau_j \dot{q}_j||,$$

where $w_{\{.\}}$ and $\alpha_{\{.\}}$ are weights to balance rewards. The reward encourages the robot to imitate the position $\hat{p}$, rotation $\hat{q}$, linear velocity $\hat{\dot{p}}$, angular velocity $\hat{\dot{q}}$, and the root height $\hat{h}$ specified by the reference motion. An energy penalty is applied to encourage smoother motion and mitigate jittering.

**Training.** Following MaskedMimic (Tessler et al., 2024; Wang et al., 2025b), the policy $\pi$ is modeled using a transformer encoder architecture, and the critic is modeled with a simple MLP. To enhance training stability and efficiency, we adopt the Reference State Initialization (RSI) and Early Termination (ET) strategies introduced in DeepMimic (Peng et al., 2018). Specifically, at the beginning of each episode, the initial state is sampled from the reference motion: with probability 10% from the first frame, and otherwise uniformly along the trajectory. An episode is terminated early if any joint position deviates by more than 0.5 meters from the reference in world coordinates. To better compare how the reconstructed assets from different methods perform in simulation, we train a separate control policy for each motion clip. We utilize Isaac Gym (Makoviychuk et al., 2021) to simulate all the environments with a simulation frequency of 120Hz. Policies operate at 30Hz and are optimized with Proximal Policy Optimization (PPO) (Schulman et al., 2017), using generalized advantage estimator (GAE) (Schulman et al., 2015).

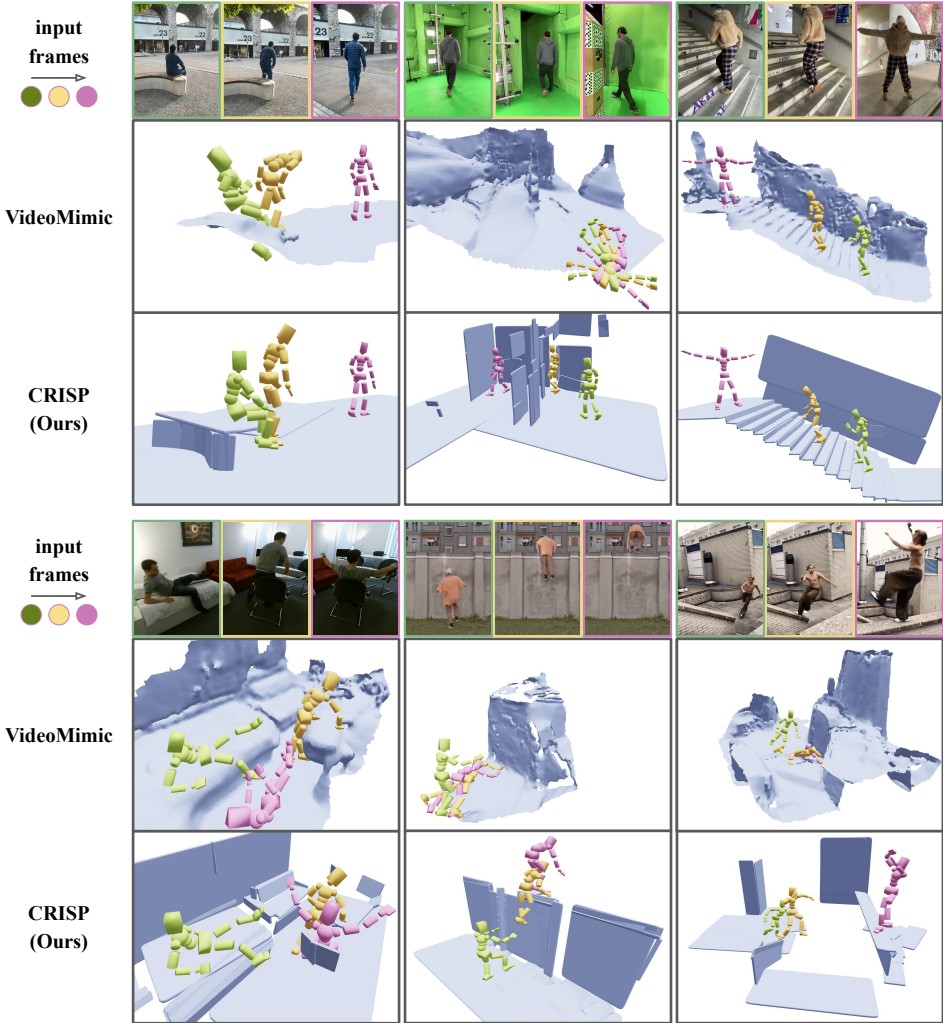

Figure 4: **Qualitative comparison.** We compare VideoMimic with CRISP (ours) on six sequences (2×3). For each sequence, the top row lists sequential input frames (green → yellow → pink). In the bottom row, we show the scene reconstructions (blue) from VideoMimic and CRISP, along with a simulated *dynamic* motion rollout at the corresponding frame. To mimic the behavior in input frames, the agents depend on accurate scene geometry for faithful physics feedback. Thus, non-physically plausible reconstruction of VideoMimic will cause simulation failure: the agent often: (a) suffers penetrations and gets stuck in 'ghost surfaces'; (b) bounces off surfaces protruding out of the ground; (c) suffers contact errors from over-smoothed structures that degrade motion recovery; (d) collides with scene artifacts; (e) gets trapped by bumpy terrains and (f) gets trapped in local dents. In contrast, CRISP produces simulation-ready assets that preserve contact-faithful human-scene interactions, enabling stable policy rollouts in diverse, complex terrain both in indoor and outdoor scenarios. Please see the project page for interactive and in-the-wild examples.

## 4 EXPERIMENTS

We evaluate CRISP on its ability to bridge monocular reconstruction and physics-based simulation along three dimensions: (i) world-grounded HMR accuracy, assessing the quality of global human motion estimation; (ii) human–scene interaction fidelity, measuring the geometric and physical consistency between reconstructed humans and environments; and (iii) RL-based motion tracking performance, evaluating whether the reconstructed assets can be used for downstream reinforcement learning. Importantly, both the human and scene reconstructions play a central role in reliable real-to-sim transfer: low-quality geometry often results in unstable contacts and poor RL convergence, whereas accurate human reconstruction supports robust tracking and physically consistent behavior.

Table 1: **Quantitative comparison.** We evaluate the humanoid policy success rates (Success, ↑), average simulation throughput (FPS ↑), reconstruction quality ($\mathcal{CD}_{bi}$ ↓ and $\mathcal{CD}_{one}$ ↓), non-penetration accuracy (Non-Pene. ↑), and HMR quality (W-MPJPE$_{100}$,↓ and WA-MPJPE$_{100}$,↓) compared to VideoMimic, as well as variants of our method with TSDF, NKSR, and Planar Primitive geometry representations. Compared to VideoMimic, our method has nearly half the chamfer distance error, and experiences much higher humanoid policy success rates. Notably, the reconstruction artifacts of VideoMimic often lead to catastrophic failures during simulation and policy rollout (Fig. 4). Compared to strongest dense mesh baseline NKSR, our 2-way chamfer error is slightly larger because our reconstructions are less complete, but our low 1-way chamfer (Recon→GT) error reveals that our planar primitives consistently lie near the ground-truth.

| Method | RL | Success ↑ | FPS ↑ | PROX(11) | | | | EMDB(20) | |
| --- | --- | --- | --- | --- | --- | --- | --- | --- | --- |
| | | | | Success | $\mathcal{CD}_{bi}$ ↓ | $\mathcal{CD}_{one}$ ↓ | **Non-Pene.** ↑ | Success ↑ | W-MPJPE$_{100}$ ↓ (WA-) |
| VideoMimic | ✗ | — | — | — | 0.337 | 0.311 | 0.928 | — | 521.09 (110.64) |
| Ours | ✗ | — | — | — | 0.187 | **0.174** | 0.909 | — | 179.84 (78.16) |
| VideoMimic | ✓ | 44.8% | 16K | 27.3% | 0.337 | 0.311 | 0.906 | 50.0% | 505.31 (145.23) |
| Ours (TSDF) | ✓ | 75.9% | 15K | 72.7% | 0.178 | 0.222 | 0.925 | 77.8% | 197.77 (75.62) |
| Ours (NKSR) | ✓ | 79.3% | 16K | **90.9%** | **0.163** | 0.187 | 0.937 | 75.0% | 185.00 (74.77) |
| **Ours (Planar)** | ✓ | **93.1%** | **23K** | **90.9%** | 0.187 | **0.174** | **0.947** | **93.8%** | 175.93 (70.60) |

## 4.1 Experiment Setup

**Datasets.** We conduct experiments on EMDB (Kaufmann et al., 2023) and PROX (Hassan et al., 2019). EMDB provides ground-truth global human motion without paired scene geometry. Following prior work (Shin et al., 2024; Wang et al., 2024a), we use the EMDB-2 subset with 21 sequences (4 indoor, 17 outdoor). PROX contains pseudo-ground-truth human motion paired with 3D scene scans in 12 indoor settings. As the pseudo-ground-truth annotations in PROX are known to be noisy, most prior work does not report direct comparisons on motion accuracy. In line with this practice, we use PROX primarily for evaluating human–scene interaction fidelity. We standardize each video to 600 frames(20s) on average. During meshification, points beyond the 95th-percentile depth (far from the camera) or farther than 2.5m from the pelvis are treated as non-contact and filtered out.

**Baselines.** We evaluate our method against two types of baselines: alternative geometry reconstruction methods and alternative human motion recovery (HMR) methods. For both, we use the same RL-based motion tracking framework for comparison of downstream simulation performance. We also extract reference motion and geometry from CRISP and VideoMimic (Allshire et al., 2025) and send them into the same tracking and RL training pipeline for fair system-to-system comparison.

*Geometry.* We consider several common scene reconstruction pipelines: (i) meshes reconstructed from TSDF fusion using VDBFusion (Vizzo et al., 2022) and Marching Cubes (Lorensen & Cline, 1987), (ii) point clouds reconstructed via NKSR (Huang et al., 2023), and (iii) dense mesh reconstructions adopted by VideoMimic (Allshire et al., 2025). These baselines are compared to our planar primitive representation, which produces lightweight, simulation-ready geometry.

*World-grounded HMR.* For global human motion recovery, we compare against prior state-of-the-art methods, including GVHMR (Shen et al., 2024), TRAM (Wang et al., 2024a), WHAM (Shin et al., 2024), and VideoMimic (Allshire et al., 2025). Beyond raw HMR output from VideoMimic and CRISP (ours), we also include the rollout motion from trained RL policies for comparison.

By combining these baselines with the same RL-based physics refinement, we obtain a fair assessment of how different human and scene models impact the final real-to-sim performance. Notably, when both human and scene reconstructions are high quality, RL refinement produces motions that are not only physically stable but also closely resemble those observed in the input video.

**Metrics.** We evaluate all methods using three categories of metrics:

*World-grounded HMR quality.* Following prior work (Shin et al., 2024; Wang et al., 2024a), we split sequences into 100-frame segments and align the predicted motion with ground truth, using either the first two frames or the full segment. We report WA-MPJPE$_{100}$ and W-MPJPE$_{100}$ (mean per-joint position error in millimeters), root translational error (RTE, normalized by trajectory length), as well as temporal smoothness metrics such as jitter and acceleration error.

*Human–scene interaction fidelity.* To measure geometric alignment, we compute Chamfer Distance between reconstructed and ground-truth scenes, including bi-directional and one-way variants

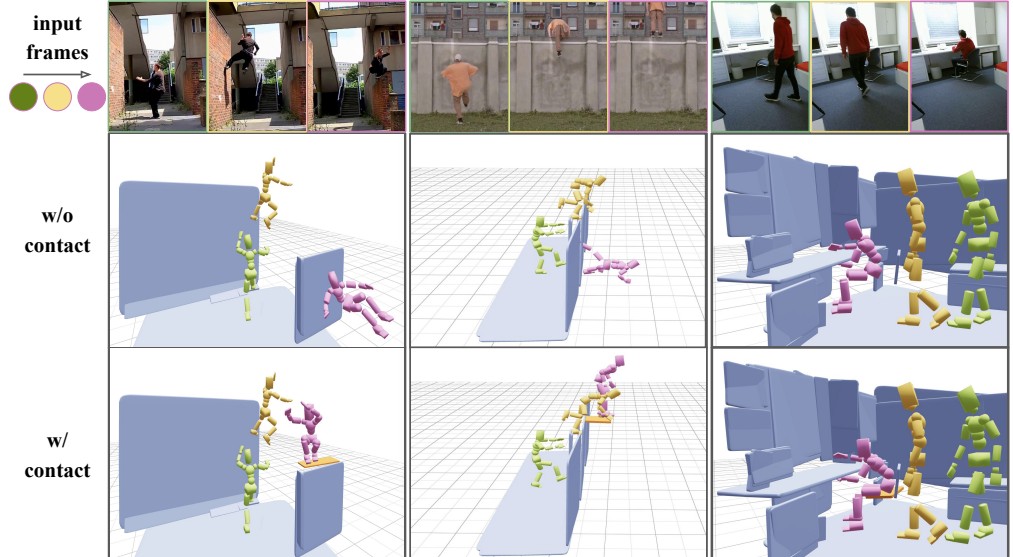

Figure 5: **Ablation on contact-guided scene completion.** We evaluate impact of contact-guided scene completion on three sequences (three columns). For each sequence, the top row lists sequential input frames (green → yellow → pink). Below, we overlay the dynamics rollout at the corresponding time step on the reconstructed scene geometry (blue). The middle row ("w/o contact") reconstruction omits occluded support surfaces (e.g. platforms), causing the humanoid to fall or move unnaturally and leading to physically implausible rollouts. In contrast, the bottom row ("w/ contact") includes additional support geometry (orange), producing stable simulations and motion that better tracks the reference human motion. Please see video on website for more details.

(Recon→GT and GT→Recon). Precision is reflected in low Recon→GT error (every reconstructed surface is supported by ground truth), while completeness is reflected in GT→Recon error. We also report the non-penetration score (Non-Pene) following PLACE (Zhang et al., 2020a), which quantifies how often human meshes intersect with scene geometry.

*RL-based motion tracking performance.* We assess whether the reconstructed assets can be used for reinforcement learning. We report two metrics: task success rate and training throughput (FPS). Success rate is defined as the percentage of episodes in which the robot's joints remain within 0.5 meters of the reference motion throughout the trajectory. FPS measures the overall system throughput, including both environment simulation and policy learning steps, reflecting the efficiency of RL training. Strong performance of CRISP shows it not only reconstructs geometry and motion, but also produces assets that enable controllable, video-faithful simulation.

## 4.2 OVERALL REAL-TO-SIM PERFORMANCE

Table 1 summarizes the overall quantitative results for reconstruction fidelity, contact quality, and downstream RL training. Compared to the concurrent VideoMimic pipeline, our method achieves substantially higher RL success rates and simulation throughput: while VideoMimic attains only 44.8% success with 16K FPS, our planar primitive representation reaches 93.1% success with 23K FPS, significantly improving both reliability and efficiency.

We further ablate the effect of scene representation in Table 1. Replacing VideoMimic's dense mesh with a TSDF mesh already improves RL success, but the TSDF geometry still exhibits over-smoothing and duplicated structures, which reduce contact precision. Using NKSR sharpens the reconstructed surfaces, yielding stronger non-penetration and lower Chamfer distance. Our planar primitives push this trend further: they achieve the lowest *one-sided* Chamfer ($CD_{one}$, Recon→GT) while incurring a slightly worse *bidirectional* Chamfer $CD_{bi}$. This gap is expected, since $CD_{bi}$ includes the GT→Recon term that penalizes missing fine-grained structures, which are typically non-contact regions in human–scene interaction. In contrast, the one-sided Recon→GT error directly reflects how accurate the reconstructed geometry is where it *does* exist. In simulation, missing tiny non-contact details is largely harmless, whereas extra noisy geometry produces unwanted contact artifacts that can destabilize policy rollouts. Consequently, despite a slightly worse $CD_{bi}$, our pla-

Table 2: **World–grounded HMR Evaluation on __EMDB__.** We report WA-MPJPE$_{100}$, W-MPJPE$_{100}$, Root-Translational Error (RTE), Jitter (J), and Acceleration error (ACCEL). Our method outperforms all baselines after RL motion refinement.

| Method | RL | WA-MPJPE$_{100}\downarrow$ | W-MPJPE$_{100}\downarrow$ | RTE$\downarrow$ | Jitter$\downarrow$ | ACCEL$\downarrow$ |
|---|---|---|---|---|---|---|
| WHAM | ✗ | 98.45 | 267.53 | 3.30 | 22.57 | 5.21 |
| TRAM | ✗ | 83.61 | 249.50 | 1.93 | 24.00 | 4.82 |
| GVHMR | ✗ | 74.80 | 200.71 | 1.90 | 15.50 | 4.39 |
| VideoMimic | ✗ | 110.64 | 521.09 | 2.12 | 9.29 | 4.65 |
| VideoMimic | ✓ | 145.24 | 505.32 | 3.00 | 8.34 | 4.17 |
| **Ours** | ✗ | 78.16 | 179.84 | **1.88** | 13.04 | 4.59 |
| **Ours** | ✓ | **70.60** | **175.93** | 1.90 | **8.14** | **4.10** |

nar representation yields the highest non-penetration score and the best RL success rate, making it particularly well-suited to our contact-rich setting. One remaining limitation is that planar decomposition can leave small gaps between neighboring primitives, so the reconstructions may appear visually incomplete; however, we observe that these gaps do not affect simulation quality because key support and contact surfaces are always modeled.

We present a qualitative comparison with VideoMimic in Fig. 4, including both the reconstructed scenes and the outcomes after RL-based simulation. Additional visualizations are available on our project website. These results reveal a clear trend: lower-quality human or scene reconstructions propagate errors into the physics loop, leading to unstable contacts and reduced RL tracking success, whereas our higher-fidelity, simulation-optimized representations enable robust policy learning.

### 4.3 GLOBAL HUMAN MOTION ESTIMATION

Since our goal is to build a real-to-sim pipeline that remains faithful to the original video, it is essential to validate the accuracy of human motion recovery. Table 2 reports the global motion estimation results on EMDB using world-grounded metrics. Our method significantly outperforms prior approaches in both joint accuracy and trajectory stability. Without RL refinement, our model already achieves competitive WA-MPJPE$_{100}$ and W-MPJPE$_{100}$, comparable to GVHMR and better than WHAM, TRAM, and VideoMimic. With RL-based motion tracking, errors are further reduced to 70.60 mm and 175.93 mm respectively, yielding the best overall accuracy among all methods.

In addition to joint-level metrics, our method also improves global trajectory consistency, with the lowest root translational error and substantially reduced temporal jitter (8.14) compared to previous methods. This indicates that reinforcement learning not only reduces pose drift but also stabilizes temporal dynamics, producing smoother and more physically consistent motion. By contrast, VideoMimic exhibits large joint errors ($> 500$ mm) and high drift, showing that low-quality reconstructions severely degrade world-grounded motion estimation.

### 4.4 ABLATION STUDY ON CONTACT CUES

To assess the role of contact cues, we conduct an ablation study with and without our VLM-based contact prediction. As shown in Table 3, explicitly incorporating contact signals improves the geometric alignment of reconstructed scenes, yielding lower Chamfer Distance in both GT→Recon and Recon→GT directions. The inclusion of contact priors helps refine surfaces near human interaction regions, leading to more faithful reconstructions of support structures. In Figure 5, we illustrate several qualitative examples showing that without these contact-guided planar reconstructions, downstream RL policies often struggle to finish the entire motion.

## 5 CONCLUSION

We present CRISP, a real-to-sim pipeline that transforms unconstrained monocular human videos into simulation-ready assets, enabling physically valid and natural human–scene interactions. Our planar fitting yields compact, high-quality, and simulation-ready convex scene primitives, and our contact-guided reconstruction strategy allows us to recover occluded surfaces, while reinforcement learning validates and refines the reconstructed interactions in simulation. Extensive experiments demonstrate that CRISP substantially improves world-grounded HMR accuracy, human–scene interaction fidelity, and downstream RL training performance.

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

## A  APPENDIX

**The Use of Large Language Models (LLMs).**  The authors confirm that LLMs are used to polish paper writing in abstract and introduction sections. Some captions use output by LLMs.

**Reproducibility statement.**  We have tested all code on publicly-available online videos and have validated its success. All experiments were conducted on standard benchmarks and additional publicly available internet videos. Code and data will be open-sourced upon acceptance.

## B  DETAILS OF PLANAR FITTING.

---

**Algorithm 1:** Planar Fitting from $[T, N, 3]$ to $[M, \mathbf{R}, \mathbf{t}, \mathbf{S}]$ with $\Pi \in \{0, 1\}^{[NT, M]}$

---

**Input:** Per-frame points $\{P_t\}_{t=1}^T$ with $P_t \in \mathbb{R}^{N \times 3}$; optical flows $\{\Phi_{i \to j}\}$; covisibility masks $\{\mathcal{C}_{i,j}\}$.

**Output:** $\mathbf{R} \in \mathbb{R}^{M \times 3 \times 3}, \mathbf{t} \in \mathbb{R}^{M \times 3}, \mathbf{S} \in \mathbb{R}^{M \times 3}$, and $\Pi \in \{0, 1\}^{[NT, M]}$.

1  **(1) Per-frame segmentation**
2  **for** $t = 1, \ldots, T$ **do**
3      (a) *Estimate normals from points*: $P_t [N, 3] \to N_t [N, 3]$.
4      (b) *Cluster normals into K groups*: $N_t [N, 3] \xrightarrow{\text{KMeans}} y_t [N]$ (labels in $\{1, \ldots, K\}$).
5      (c) *Spatial clustering*: For each $k$, take $X_k = \{p \in P_t : y_t(p) = k\} [N_k, 3] \xrightarrow{\text{DBSCAN}} \{\mathcal{S}_t\}$.
6  **(2) Cross-frame association (set view)**
7  **for** *each pair* $(i, j)$ *with* $\Phi_{i \to j}$ *and* $\mathcal{C}_{i,j}$ **do**
8      (a) *Warp*: For each segment $a \in \mathcal{S}_i$, warp $\mathcal{S}_{i,a}$ from frame $i$ to frame $j$ by $\Phi_{i \to j}$.
9      (b) *Score & Link*: Let $A = \mathcal{S}_i$, $B = \mathcal{S}_j$ with $a \in A, b \in B$. For each $b \in \mathcal{S}_j$, compute overlap ratio $\rho_{ab}$ between the warped $\mathcal{S}_{i,a}$ and $\mathcal{S}_{j,b}$, and normal cosine $\gamma_{ab} = \langle \bar{n}_{i,a}, \bar{n}_{j,b} \rangle$. Connect $(i, a) \leftrightarrow (j, b)$ if both scores above fixed thresholds. Set $X = A \cap B$ on the warped domain (accepted overlaps) and produce $A + B - X$ segments instead.
10     (d) *Unify across pairs*: Aggregate all $(A + B - X)$ over time $\Rightarrow M$ global planar groups.
     `// we have found points-to-plane correspondence`
11  **(3) Primitive fitting (global → primitive)**
12  **for** $m = 1, \ldots, M$ **do**
13      (a) *Plane fitting (RANSAC)*: $X_m = \bigcup \{p \in P_t\} [N_m, 3] \to (n, c)$ and inliers.
14      (b) *Pose*: Project inliers to the plane; fit a min-area rectangle to get in-plane axes $(x, y)$; set $\mathbf{R}_m = [x \ y \ n]$ (right-handed). *(Defer center update to (c))*
15      (c) *Size & Center*: Set $\mathbf{S}_m = [S_x, S_y, S_z]$ from in-plane coverage and normal-direction spread; **then set**
$$\Delta = \tfrac{1}{2} S_z n \quad \text{and} \quad \mathbf{t}_m = c + \Delta.$$
16      (d) *Split (optional)*: If the footprint is not rectangular, split along the principal in-plane axis and refit on each part.
17  **(4) Contact-guided hallucination (opt.)** Repeat the above fit on predicted contact points (world coords) to augment planes, with $S_z \geq 0.05$ m clamp.
18  **return** $(\mathbf{R}, \mathbf{t}, \mathbf{S}, \Pi)$.

---

## C  CONTACT-GUIDED SCENE COMPLETION: RELIABILITY ANALYSIS

Our contact-guided completion uses temporal-kinematic filtering to suppress false-positive contact predictions. Specifically, we retain windows of length $L$ with consistently high contact confidence (threshold $\tau$) and low body speed (threshold $\nu$), then choose the most stationary frame per window. We find this reduces premature "imminent contact" false positives and improves the stability.
**When does contact guidance help?**  Contact-guided completion is most beneficial when support surfaces are occluded (e.g., chair seats, stair treads, platforms). In these cases, completing even a small missing support region can prevent catastrophic falls and significantly increase rollout success. We use RL-based motion tracking to *validate* and *physically ground* the recovered human motion within the reconstructed scene. Concretely, RL optimizes a control policy that tracks the recovered

Table 3: **Ablation on Contact (no RL-refine).** Green ✓ = enabled, red ✗ = disabled. For PROX, higher Success / Non-Pene and lower CD are better; for PKR, we report Success only. CD (two-way) is the symmetric Chamfer, while GT→Recon and Recon→GT are the directional terms. Contact modeling on the PROX dataset does not increase the success rate because most sequences involve sitting; when the seat is missing, the motion simply switches to a squatting pose. However, contact modeling still improves physical plausibility quantitatively as shown in Fig. 5.

| Method | Contact? | PROX | | | | |
|---|---|---|---|---|---|---|
| | | Success ↑ | CD (two-way) ↓ | $CD_{GT \to Recon}$ ↓ | $CD_{Recon \to GT}$ ↓ | Non-Pene ↑ |
| VideoMimic | ✗ | 27.3% | 0.337 | 0.3625 | 0.3114 | 0.928 |
| Ours | ✗ | 90.9% | 0.193 | 0.211 | 0.175 | 0.947 |
| Ours | ✓ | 90.9% | 0.187 | 0.199 | 0.173 | 0.947 |

reference motion while respecting the scene's contact constraints. This process refines the *simulated motion rollouts* (e.g., reducing drift/jitter and eliminating physically implausible penetrations), and serves as an end-to-end test of whether the reconstructed assets are *simulation-ready*. Importantly, in the current implementation RL does not directly modify the reconstructed scene primitives; rather, it uses them as collision geometry and assesses whether the motion can be executed stably.

**Failure pattern of contact guidance.** Although we empirically find that our contact completion method can improve overall reconstruction quality by hallucinating support surfaces, we find that the estimated contact-augmented planar primitives are occasionally inaccurate. We use an HMR network to infer 3D human pose and shape, then run an off-the-shelf contact prediction module to infer binary contact masks on the canonical SMPL mesh. This paradigm highly depends on the accuracy of HMR and contact estimation: if the human mesh drifts in world coordinates, the resulting contact-augmented planar primitives would drift as well. Most of the time, this appears as slightly tilted planar primitives with a small amount number of offset translation. For future work, improving the planar fitting or post-processing the contact-completed geometry to make it more scene-grounded is a promising direction.

# D LIMITATIONS AND FUTURE WORK

We summarize several failure cases:

**(1) Inherent limitations of planar primitives.** While we empirically show that curved or round objects can often be approximated reasonably well (See our website for results), highly curved or organic shapes may appear faceted or under-fitted. Although this limitation does not significantly reduce locomotion success rates, we believe that integrating our method with more expressive convex primitives (e.g., superquadrics) could further improve fine-grained shape details for curved surfaces.

**(2) Unable to model fluid and deformable objects.** Our method assumes a static, rigid scene geometry. Thus, we are not able to faithfully model fluids (e.g., sand in a desert, water in a lake) or deformable objects (e.g., folded cloth, soft cushions when in large deformations). Nonetheless, the existing codebase could be extended to dynamic rigid objects or scenes by incorporating time-varying primitives.

**(3) Unable to model dynamic objects.** CRISP is limited to static human-scene interaction, and does not currently support dynamic scenes or moving objects. The RL training is for locomotion only and lacks loco-manipulation ability as in existing pipelines Weng et al. (2025); Kuang et al. (2025).

**Future work**. Extending CRISP with more expressive primitives (e.g., superquadrics or other parametric shapes) and supporting dynamic human-object interaction handling are exciting directions for future work and could further improve local geometric fidelity, especially for modeling of complex scene and coupling affordance prediction (Wan et al., 2025) for manipulation interactions.

# E RUNTIME ANALYSIS

We test CRISP on a 300-frame (10s) video with resolution of 1440*1920 on a single RTX A6000 GPU. We report the cost for each component of our pipeline in Table 4. Meanwhile, VideoMimic requires 1282.94s on the same machine to prepare geometry and reference motion before RL. From the result, we can conclude that the main computational bottleneck is Visual SLAM (i.e. MegaSAM) and inferring the feed-forward priors that are required (monocular depth and flow estimation). Our planar fitting algorithm is lightweight, meaning that the core geometric step (from points to planar

Table 4: Runtime and memory breakdown of CRISP on a 300-frame (10 s) video at $1440 \times 1920$ resolution using a single NVIDIA RTX A6000.

| Module | Runtime (s) | Runtime (min) | VRAM (MiB) | Proportion (%) |
|---|---|---|---|---|
| 1. Prior preparation | 297.33 | 4.96 | 4,944 | 32.3 |
| 2. Visual SLAM | 518.18 | 8.64 | 11,936 | 56.3 |
| 3. HMR (GVHMR) | 30.51 | 0.51 | 4,940 | 3.3 |
| 4. Planar fitting | 74.97 | 1.25 | – | 8.1 |
| **Total** | **920.99** | **15.35** | **11,936** | **100.0** |

primitives) can be run in real time on incoming frames. When coupled with a real-time RGB-D SLAM and detection system (Wang et al., 2024b), we expect that the pipeline can operate in a real-time fashion, using sensor depth and flows obtained from relative camera motion.

## F  SCENE-AWARE POLICY.

Beyond the scene-blind baseline in the paper, we implemented and evaluated a scene-aware variant that takes geometry information at runtime. Specifically, we downsample a sparse number [N=2.8k] of 3D points from nearby planar primitives, represent these points in the humanoid's local frame, and apply a PointNet to extract global features which serve as additional input tokens for the transformer policy (scene-blind: proprioception + target poses; scene-aware: proprioception + target poses + point cloud). This scene-aware policy achieves higher success rates and better obstacle avoidance in some sequences; we show the comparisons on the project website. However, the main goal of our work is not to train a controller that can compensate for imperfect reconstructions, but to test how faithfully our reconstructed scenes and motions support direct simulation ("real-to-sim"). A scene-aware policy can learn to avoid obstacles and route around reconstruction errors, but doing so reduces its sensitivity to reconstruction quality and makes it harder to reveal underlying reconstruction issues. Our experiments show that once geometry and motion capture are accurate, the blind policy already behaves robustly; scene-aware inputs provide an optional extra performance gain for deployment (e.g., when targeting sim-to-real), but they are not required to substantiate the central claims of the paper.

## G  SEAMLESS TRANSFER TO CURRENT SIM-TO-REAL PIPELINE.

We believe that CRISP can be readily integrated into existing sim-to-real robot pipelines; for example, it is a drop-in replacement for stages 1 and 2 of VideoMimic's 4-stage pipeline (1. MoCap pretraining, 2. tracking reconstructed video motions, 3. distillation, 4. RL finetuning).

## H  RL TRAINING DETAILS

For RL training, we use a discount factor of $\gamma = 0.99$ and apply GAE (Schulman et al., 2015) with $\tau = 0.95$. The policy is optimized with a learning rate of $2 \times 10^{-5}$. Training is conducted in parallel with 2048 environments, and each update step uses a batch size of 8192 for the policy and critic.

The tracking reward is defined with the following weights:

$$w_p = 2.5, \quad w_r = 1.5, \quad w_v = 0.5, \quad w_\omega = 0.5, \quad w_h = 1, \quad w_e = 0.001,$$
$$\alpha_p = 1.5, \quad \alpha_r = 0.3, \quad \alpha_v = 0.12, \quad \alpha_\omega = 0.05, \quad \alpha_h = 20. \tag{1}$$

For the policy architecture, we adopt a transformer encoder with a latent dimension of 256, feedforward size of 512, two layers, and two self-attention heads. The critic is implemented as a multi-layer perceptron with hidden sizes [1024, 512].

We show the learning curves of our method and VideoMimic on each motion clip in Figure 6. Our reconstructed assets allow the agent to achieve higher rewards more quickly, indicating faster convergence and better overall tracking quality. In addition, we visualize the torque reward over training steps in Figure 7. Compared to our method, VideoMimic consistently requires the robot to exert

higher torques to complete the same motions. This reflects the lower quality of the human–scene assets produced by VideoMimic, which introduce unwanted geometry artifacts and unstable supports, thereby forcing the policy to compensate with excessive control effort.

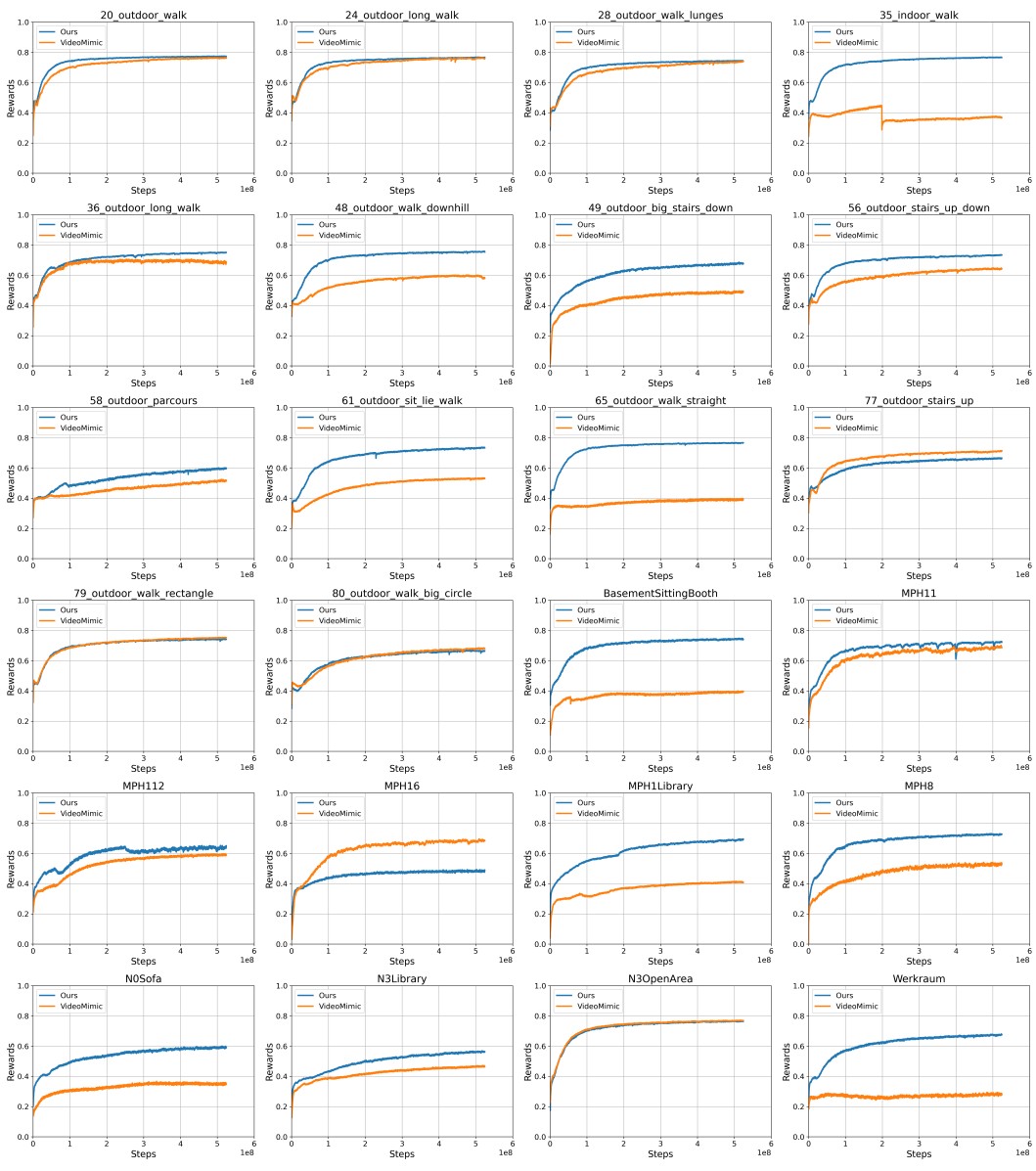

Figure 6: **Reward curve.**

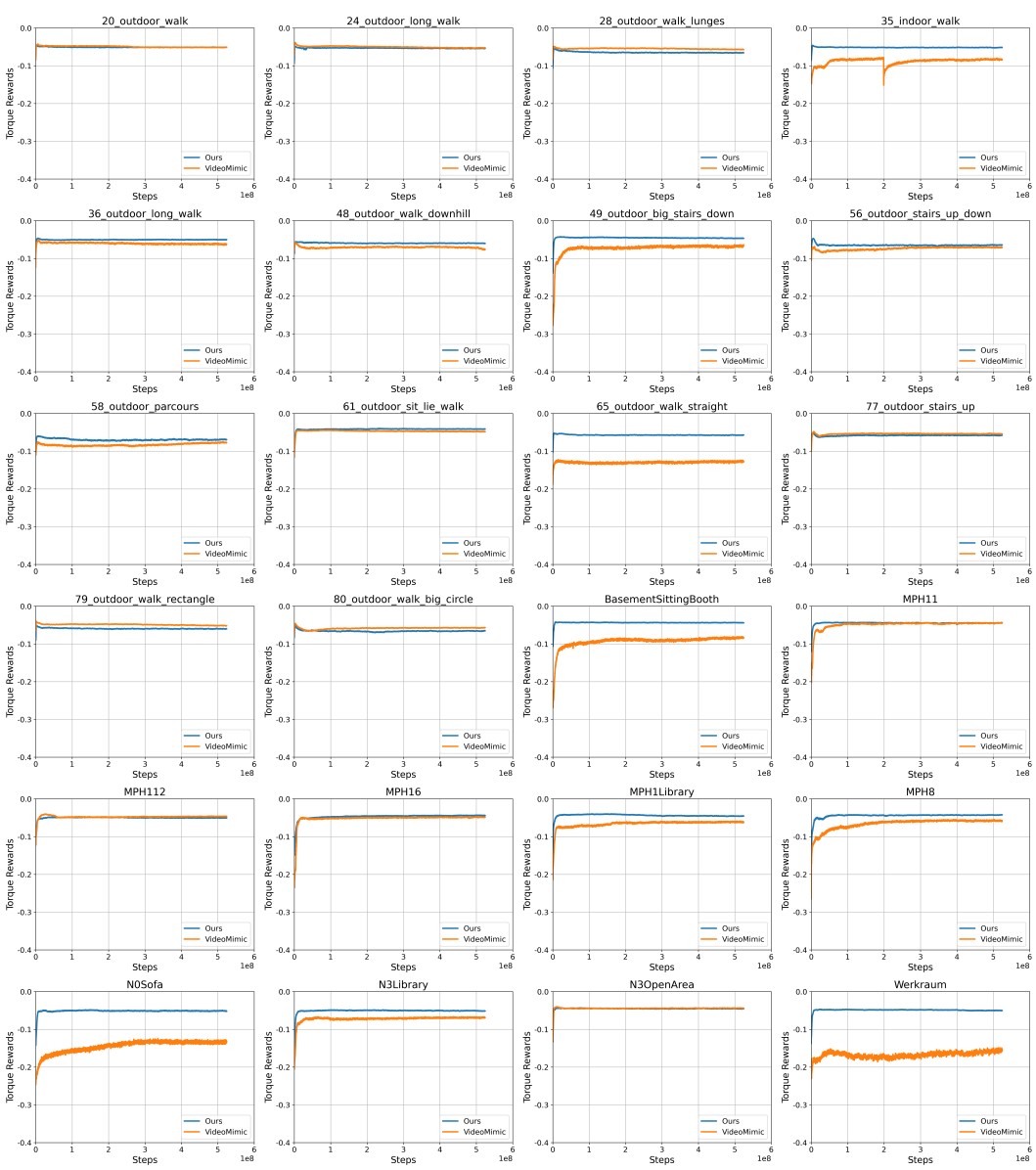

Figure 7: **Torque reward curve.**

