# OpenReview forum: "Contact-guided Real2Sim from Monocular Video with Planar Scene Primitives"
_ICLR.cc/2026/Conference — ICLR 2026 Poster_

### Official Review · Reviewer_8a6k · 2025-10-26

**Soundness:** 2
**Presentation:** 2
**Contribution:** 1
**Rating:** 2
**Confidence:** 5

**Summary:**

The paper introduces CRISP, a method for recovering simulatable 3D human motion and scene geometry from monocular video. Its key point is the use of compact planar primitives for scene representation, which are more efficient and robust for physics simulation than traditional dense meshes. CRISP also employs contact-guided modeling to reconstruct occluded interaction surfaces. By validating the reconstruction through Reinforcement Learning (RL), the method achieves a high real-to-sim success rate and reduces motion tracking failure rates, while speeding up RL simulation throughput. This makes the "vid2sim" pipeline practical for embodied AI and robotics.

**Strengths:**

The paper's strengths lie in its high empirical quality and practical value for the Real-to-Sim pipeline. It replacing noisy dense meshes with convex planar primitives for simulation efficiency. This engineering choice, combined with clever contact-guided scene completion, results in a highly robust system that achieves a  good reduction in motion tracking failures and an acceleration of RL training throughput.

**Weaknesses:**

The primary weakness of CRISP is its limited conceptual novelty, functioning largely as a highly effective "set of practical engineering approaches" that integrates existing components (HMR, 4D reconstruction, contact prediction) rather than introducing a new framework. Specifically, the use of planar primitives, while practical, relies on a simple, non-learned clustering pipeline that may struggle with complex, non-planar geometries (e.g., curved furniture, organic shapes), limiting its generalizability beyond the "planar-world assumption." The Contact-Guided Scene Completion is a heuristic that depends on pre-trained vision-language models for interaction detection, which introduces a dependency on external, potentially brittle, classifiers. The two core designs—reliance on planar primitives and contact-guided scene completion—may boost computational efficiency but fail to address the key challenges in scene reconstruction, leaving the work unable to grasp the true hurdles of such tasks.
In addition, the experimental comparison in Figure 2 is not fair to the baseline method (VideoMimic). The authors represent the scene using compact planar primitives, which, while simplifying the optimization for motion tracking, lead to significant scene distortion. This representation creates a larger sim-to-real gap for the subsequent humanoid tracking task. For a more equitable comparison and to better prepare the method for real-world deployment, the authors should consider investigating learning a motion prior from these compact scene interactions.

**Questions:**

no.

---

> ### Author Response · Authors · 2025-11-21
> **Response to reviewer 8a6k (1/2)**
>
> - Q: limited conceptual novelty integrating existing components.
> - A: We addressed it in the General Response.
>
> - Q: The Contact-Guided Scene Completion is a heuristic that depends on pre-trained vision-language models for interaction detection, which introduces a dependency on external, potentially brittle, classifiers.
> - A: We admit that CRISP relies on off-the-shelf vision–language models for interaction cues as a heuristic procedure. However, this is a principled heuristic with clear design intuition, that contacts provide strong signals about which surfaces must be geometrically and physically plausible. Moreover, the reviewer provides no empirical or logical evidence that these models are “brittle” in our setting. On the contrary, we observe stable interaction cues and consistent improvements over baselines that do not use contact guidance (we refer to Sec. B in Appendix) and downstream policy performance over baselines without contact guidance.
>
> - Q: The two core designs—reliance on planar primitives and contact-guided scene completion—may boost computational efficiency but fail to address the key challenges in scene reconstruction, leaving the work unable to grasp the true hurdles of such tasks.
> - A: We interpret the “key challenges in scene reconstruction for real‑to‑sim” and the “true hurdles of such tasks” as three concrete problems: (i) recovering metrically accurate geometry from monocular video, this requires finding the metric signal from in-the-wild video to solve scale-ambiguity of SLAM. (ii) gap between visual reconstruction and simulation performance: directly using standard visual reconstructions (e.g., raw meshes) often produces scenes that look reasonable but fail in physics. (iii) handling persistent occlusions by hallucinating physically necessary but unseen surfaces. Contact surfaces often stay occluded or only partially visible. A purely geometric pipeline cannot reconstruct them reliably, which again breaks real‑to‑sim. CRISP explicitly tackles all three: (i) we use the metric SMPL model as a clue to align scene reconstruction with HMR, which solves scale ambiguity. (ii) we use a simulation‑ready primitive representation, which yields 2–3× higher tracking success rates than concurrent work in the same environments. (iii) we predict human–scene contacts and use them as cues to hallucinate missing support surfaces in occluded regions. If the reviewer has additional doubts in mind, we would be grateful for concrete explanations so that we can better understand the key challenges and the true hurdles of such tasks to grasp.
>
> - Q: limited generalizability beyond the "planar-world assumption."
> - A: The goal of our work is not general 3D reconstruction, but to obtain simulation‑ready geometry for locomotion interactions (e.g., sitting, climbing stairs, parkour). In these settings, we find planar primitives are a suitable approximation, as most contacts occur on surfaces that are well approximated by planes (floors, steps, tabletops, furniture). Planar primitives are flexible enough to fit curved structures: as shown in the first example of Figure 3, even strongly curved geometry such as the bench cushion can be modeled by a set of planar patches that closely follow its arc. Although highly curved or organic objects (e.g., wheels of skateboards, soft cushions under large deformation) may appear faceted or under‑fitted, this mainly affects fine‑grained shape details rather than the contact surface for the tasks we target. We empirically do not observe failures in locomotion due to this approximation. Extending CRISP with more expressive primitives (e.g., superquadrics or other parametric shapes) is an exciting direction for future work and could further improve local geometric fidelity, especially for manipulation interactions.

---

> ### Author Response · Authors · 2025-11-21
> **Response to reviewer 8a6k (2/2)**
>
> - Q: In addition, the experimental comparison in Figure 2 is not fair to the baseline method (VideoMimic). The authors represent the scene using compact planar primitives, which, while simplifying the optimization for motion tracking, lead to significant scene distortion. This representation creates a larger sim-to-real gap for the subsequent humanoid tracking task. For a more equitable comparison and to better prepare the method for real-world deployment, the authors should consider investigating learning a motion prior from these compact scene interactions.
> - A: We respectfully disagree with the reviewer’s statements. We carefully ensured that the comparison with VideoMimic is fair. Specifically, we run the official VideoMimic code and strictly follow its pipeline to extract reference motion and geometry from the input video. Our method uses a different system to extract motion and geometry from the same video. We then fed the reference motion and geometry into the same tracking and RL training pipeline. Thus, the only difference in the comparison is the real-to-sim system used to recover motion and geometry from video [VideoMimic vs. CRISP].
>
> The goal is to obtain physically plausible human–scene interaction data from monocular videos, which has broad impact in robotics, AR/VR, and animation and is not limited to sim-to-real transfer. In this setting, the quality of the reconstructed scene strongly affects both tracking and policy learning. As shown in our project website visualizations (where reviewers can drag and play with it), VideoMimic’s reconstructions frequently suffer from missing support surfaces and bumpy geometry. These artifacts cause the agent to get stuck, penetrate the scene, or lose physical support. Such failure cases are harmful for motion tracking and severely limit scalability: VideoMimic achieves only 44.8% tracking success, while CRISP, using the same meshing pipeline (i.e., NKSR) for reconstruction, achieves 79.3% success. This suggests we fix the poor reconstruction problem of VideoMimic even before planar primitives. We also refer to Table 1 for reconstruction quality, in which all of our representation variants beat VideoMimic by a large margin.
>
> Considering sim-to-real applications, our scene-grounded, physically plausible motion reconstructed from video can serve as a rich data augmentation source. Such data from planar primitives can augment Stage 1 of VideoMimic (MoCap pretraining) by providing more diverse motion data beyond AMASS motion. Subsequently, our NKSR reconstruction can help enhance Stages 2 and 3, enabling more scalable and stable policy learning.

---

> ### Author Response · Authors · 2025-11-28
> **Rebuttal follow-up**
>
> Dear Reviewer ,
>
> We would like to thank you again for the time and effort you have dedicated to reviewing our submission.
>
> In our author response, we have tried to address all questions and major concerns raised by the reviewers, both in a general response and in point‑by‑point replies to each reviewer.
>
> We fully understand that the discussion period is busy, but we would be very grateful if you could take these clarifications into account when finalizing the review and score. Please let us know if any additional information would be helpful, we are happy to provide further details.

---

### Official Review · Reviewer_9RmM · 2025-10-30

**Soundness:** 3
**Presentation:** 3
**Contribution:** 3
**Rating:** 6
**Confidence:** 4

**Summary:**

This paper presents a framework for simulation-ready human scene reconstruction from a single video. The authors introduce a planar primitive representation to better model planar regions such as the ground and propose a contact term to improve physical plausibility. The paper is clearly written and provides several qualitative results demonstrating the effectiveness of the approach.

**Strengths:**

1. The paper is clearly written and easy to follow.

2. The idea of introducing planar primitives to constrain plane reconstruction is intuitive and practical.

3. The proposed contact term helps ensure more physically consistent interactions between humans and the environment.

4. The experiments validate the overall framework, and the inclusion of numerous demos and visualization results makes the paper more understandable and accessible.

**Weaknesses:**

1. Although I appreciate the effort in building the proposed framework, it somewhat gives the impression of a technical pack, combining many existing components rather than introducing fundamentally new concepts.

2. Table 2: The reported decimal precision is inconsistent across metrics and should be standardized for clarity.

3. Experimental results are not sufficiently comprehensive:

    - The ablation study is limited; for instance, it is unclear how the number of primitives affects reconstruction quality.

    - There is no report of the model’s runtime, computational cost, or efficiency.

    - No failure case analysis is provided.

    - Some of the demos provided by UPP do not seem to align properly.  In several examples, the input video and the output result do not match in duration or content.

4. The paper lacks a discussion of limitations, such as handling complex or irregular objects in the scene (e.g., interactions with yoga balls or sandbags).

5. There are several typos and minor grammatical issues throughout the paper, and need careful proofreading.

**Questions:**

What will be the result if the video is a human lying on a yoga ball?

**Details Of Ethics Concerns:**

This paper involves human subject videos.

---

> ### Author Response · Authors · 2025-11-21
> **Response to reviewer 9RmM**
>
> - Q: Missing result of handling complex or irregular objects in the scene (e.g., interactions with yoga balls or sandbags).
> - A: We include the mentioned sequences on the website and refer the reviewer to examples in in-the-wild 7 and in-the-wild 9. Although yoga ball / sandbag does not meet the planar-world assumption, our proposed system can still achieve a reasonable fit on the sphere-shell / deformed structures and demonstrate successful simulation.
>
> - Q: No report of the model’s runtime, computational cost, or efficiency.
> - A: We test CRISP on a 300-frame (10s) video with resolution of 1440×1920 on a single A6000. We outline our pipeline and report the cost in each module as follows:
>
> | Module               | Runtime (s) | Runtime (min) | Mem (MiB)    | Proportion (%) |
> |----------------------|------------:|--------------:|-------------:|---------------:|
> | 1. Prior preparation | 297.33      | 4.96          | 4,944 (Max)  | 32.3%          |
> | 2. Visual SLAM       | 518.18      | 8.64          | 11,936       | 56.3%          |
> | 3. HMR (GVHMR)       | 30.51       | 0.51          | 4,940        | 3.3%           |
> | 4. Planar fitting    | 74.97       | 1.25          |              | 8.1%           |
> | **Total**            | **920.99**  | **15.35**     | **11,936**   | **100.0%**     |
>
>
>
> Meanwhile, VideoMimic requires 1282.94s on the same machine to prepare geometry and reference motion before RL.
>
> - Q: No failure case analysis and lacks a discussion of limitations.
> - A: We summarize several failure cases by category.
>   - Inherent limitations of planar primitives. While we empirically show that curved or round objects can often be approximated reasonably well, highly curved or organic shapes may appear faceted or under-fitted. Although this limitation does not significantly reduce locomotion success rates, we believe that integrating our method with more expressive convex primitives (e.g., superquadrics) could further improve fine-grained shape details for curved surfaces.
>   - Unable to model fluid and deformable objects. Our method assumes a static, rigid scene geometry. Thus, we are not able to faithfully model fluids (e.g., sand in a desert, water in a lake) or deformable objects (e.g., folded cloth, soft cushions when in large deformations). Nonetheless, the existing codebase can be easily extended to dynamic rigid objects or scenes by incorporating time-varying primitives. Importantly, our assumption is consistent with many contemporary 3D scene representations and does not undermine our main claims about reconstructing rigid environments.
>
> - Q: It is unclear how the number of primitives affects reconstruction quality.
> - A: We report the number of primitives vs. reconstruction quality (Chamfer Distance) in the table below, which we will append in full ablation to the final version:
>
> | Number of Primitives | Chamfer Distance (↓) |
> |----------------------|---------------------:|
> | 332                  | 0.2008               |
> | 251                  | 0.1964               |
> | 59                   | 0.1870               |
> | 32                   | 0.2000               |
> | 15                   | 0.3756               |
>
>
>
> We observe a “U-shaped” relation between the number of primitives and reconstruction quality. Intuitively, too few primitives underfit, while too many introduce over-segmentation and spurious small planes that amplify noise.
>
> - Q: Method is more of a combination of existing components rather than fundamentally new concepts.
> - A: We addressed this in the General Response.
>
> - Q: Grammar / video alignment / inconsistent decimal precision.
> - A: We appreciate the reviewer for carefully checking and pointing this out. We have updated our website and addressed all such issues, which will be reflected in the final version.

---

> ### Author Response · Authors · 2025-11-28
> **Rebuttal follow-up**
>
> Dear Reviewer,
>
> We would like to thank you again for the time and effort you have dedicated to reviewing our submission.
>
> In our author response, we have tried to address all questions and major concerns raised by the reviewers, both in a general response and in point‑by‑point replies to each reviewer.
>
> We fully understand that the discussion period is busy, but we would be very grateful if you could take these clarifications into account when finalizing the review and score. Please let us know if any additional information would be helpful, we are happy to provide further details.

---

### Official Review · Reviewer_ZtWx · 2025-10-31

**Soundness:** 3
**Presentation:** 2
**Contribution:** 3
**Rating:** 8
**Confidence:** 4

**Summary:**

The authors introduce CRISP to recover human motion and scene geometry simultaneously from monocular videos. CRISP first fits convex planar primitives to depth-based point clouds, then addresses the occlusions with human-scene contact modeling. Finally, an RL policy is trained to simulate the reconstructed human-scene interaction.

**Strengths:**

- The overall pipeline is reasonable and practical. The idea of exploiting planar primitives achieves a balance between representativity and efficiency.

- The performance is impressive and superior to its counterparts.

- Comprehensive analyses are provided, revealing meaningful findings for the real2sim of HSI.

**Weaknesses:**

- The symbol system is a little complicated. Also, there are duplicated symbols like K in L232 and L243. Annotating them in pipeline figures, such as Figure 2, might help.

- In Table 2, the ground truth jitter should be provided for reference.

- It might be better if some failure cases could be shown.

**Questions:**

- How was K in L232 decided?

- How long does it take to recover one sample?

---

> ### Author Response · Authors · 2025-11-21
> **Response to reviewer ZtWx**
>
> – Q: The ground truth jitter should be provided for reference.
>
> – A: Thanks for the suggestion, we tested ground truth jittering is 7.985, while ours is 8.14, which is very close to ground truth movement.
>
>
> – Q: No failure case analysis and lacks a discussion of limitations.
>
> – A: We summarize several failure cases by category.
> - Inherent limitations of planar primitives. While we empirically show that curved or round objects can often be approximated reasonably well, highly curved or organic shapes may appear faceted or under-fitted. Although this limitation does not significantly reduce locomotion success rates, we believe that integrating our method with more expressive convex primitives (e.g., superquadrics) could further improve fine-grained shape details for curved surfaces.
> - Unable to model fluid and deformable objects. Our method assumes a static, rigid scene geometry. Thus, we are not able to faithfully model fluids (e.g., sand in a desert, water in a lake) or deformable objects (e.g., folded cloth, soft cushions when in large deformations). Nonetheless, the existing codebase can be easily extended to dynamic rigid objects or scenes by incorporating time-varying primitives. Importantly, our assumption is consistent with many contemporary 3D scene representations and does not undermine our main claims about reconstructing rigid environments.
>
>
> – Q: How was K in L232 decided?
>
> – A: It is an empirical hyperparameter set for all cases to 8.
>
> – Q: How long does it take to recover one sample?
>
> – A: We test CRISP on a 300-frame (10s) video with resolution of 1440*1920 on a single A6000. WE outline our pipeline and report the cost in each module as followed:
>
> | Module               | Runtime (s) | Runtime (min) | Mem (MiB)    | Proportion (%) |
> |----------------------|------------:|--------------:|-------------:|---------------:|
> | 1. Prior preparation | 297.33      | 4.96          | 4,944 (Max)  | 32.3%          |
> | 2. Visual SLAM       | 518.18      | 8.64          | 11,936       | 56.3%          |
> | 3. HMR (GVHMR)       | 30.51       | 0.51          | 4,940        | 3.3%           |
> | 4. Planar fitting    | 74.97       | 1.25          |              | 8.1%           |
> | **Total**            | **920.99**  | **15.35**     | **11,936**   | **100.0%**     |
>
>
> Meanwhile, VideoMimic requires 1282.94s on the same machine to prepare geometry and reference motion before RL.
>
> – Q: complicate symbol system in method
>
> – A: We apologise for confusion. We will update pseudo code algorithm 1 and algorithm 2 in paper for easier take at the final version.

---

> ### Author Response · Authors · 2025-11-28
> **Rebuttal follow-up**
>
> Dear Reviewer,
>
> We would like to thank you again for the time and effort you have dedicated to reviewing our submission.
>
> In our author response, we have tried to address all questions and major concerns raised by the reviewers, both in a general response and in point‑by‑point replies to each reviewer.
>
> We fully understand that the discussion period is busy, but we would be very grateful if you could take these clarifications into account when finalizing the review and score. Please let us know if any additional information would be helpful, we are happy to provide further details.

---

### Official Review · Reviewer_z8C3 · 2025-10-31

**Soundness:** 2
**Presentation:** 3
**Contribution:** 2
**Rating:** 4
**Confidence:** 3

**Summary:**

In this paper, the authors proposed to use planner primitive to approximate the real world scene "geometry".
The author uses MegaSAM to recover the global scene point cloud and uses planar primitive fitting to approximate the scene geometry.
Combined with RL training, this allows for recovering the human motion and the scene geometry from a monocular video.

**Strengths:**

1. It provides a nice and simple pipeline for recovering the human motion and the scene geometry from a monocular video.
Generating motions which handles human-scene interactions is a very challenging task
and crucial for downstream applicaitons in animation and robot loco-manipulation.

2. The paper is well-written and the results are convincing.
As can be seen in the paper,
a rich set of experiments and ablation studies are conducted to evaluate the performance of the proposed method.

3. The method appears robust across a diverse set of static scenes where the agent is vaulting, parkouring, and climbing.

4. This paper provides more robust scene reconstruction in sim compared to baselines such as VideoMimic.

**Weaknesses:**

1. The paper does not present sim2real results as in VideoMimic.
Considering that the two work are very similar in terms of the pipeline,
it would make presenting only real2sim results rather incremental and less convincing.

I would argue the fps in training is not that important as people have been exploring universal sim RL policy,
making retraining less needed.
But I think the proposed method will actually make more sense in real time sim2real setting,
where scene needs to be reconstructed on the fly and perhaps that's where the planer primitive fitting will come in handy.

2. It's hard to tell how scalable and robust the algorithm is given that only a collection of scenes from PROX and EMDB are used.
It will be very helpful if the authors could provide more experiments on more diverse scenes,
and maybe even extend to the in-the-wild videos from youtube for example.

**Questions:**

1. How fast can the scene reconstruction run and can it be done in real time?

2. Is it possible to do runtime scene reconstruction which only considers a subset of the scene geometry?

3. Is it possible to input the scene information into the policy observation and apply it in a runtime fashion?

---

> ### Author Response · Authors · 2025-11-21
> **Response to reviewer z8C3**
>
> - Q: More diverse scenes on in-the-wild videos
> - A: We have run CRISP on 10 more diverse in-the-wild videos ,including casual-captured, internet video, and even SORA-generated videos. Results are included in the updated website.
>
> - Q: Is it possible to input the scene information into the policy observation and apply it in a runtime fashion?
> - A: Yes, beyond the scene‑blind baseline in the paper, we implemented and compared the scene‑aware variant that takes geometry information at runtime. Specifically, we downsample a sparse number [N=2.8e3] of the nearest pointcloud from geometry, represent these points in the humanoid’s local frame, and apply a PointNet to extract global feature which served as a token for the transformer policy (scene‑blind: proprioception + target poses; scene‑aware: proprioception + target poses + point cloud). This scene‑aware policy achieves higher success rates and better obstacle avoidance in some sequences; we show the comparisons on the project website~\url{https://crisp-real-to-sim.github.io/ViserHolder/scene-aware-policy.html.
> However, the main goal of our work is not to train a controller that can compensate for imperfect reconstructions, but to test how faithfully our reconstructed scenes and motions support direct simulation (“real‑to‑sim”). A scene-aware policy can learn to avoid obstacles and route around reconstruction errors, but doing so reduces its sensitivity to reconstruction quality and makes it harder to reveal underlying reconstruction issues.
> Our experiments show that once geometry and motion capture are accurate, the blind policy already behaves robustly; scene‑aware inputs provide an optional extra performance gain for deployment (e.g., when targeting sim‑to‑real), but they are not required to substantiate the central claims of the paper.
>
> - Q: No sim2real comparison to VideoMimic, only real2sim is incremental.
> - A: We illustrate how videomimic is concurrent, why real2sim is important by itself, and our compatible extension to sim2real in General Response.
>
> - Q: fps in training is not that important as universal sim RL policy makes retraining less needed.
> - A: Faster training FPS of our method would lead to faster iteration either from scratch training or to be fine-tuned for new tasks and environments. As an analogy, from Instant-NGP to 3D gaussian splatting it is twice faster and is greatly appreciated by the community.
>
> - Q: Running speed of our system and potential real-time application.
> - A: We test CRISP on a 300-frame (10s) video with resolution of 1440*1920 on a single A6000. WE outline our pipeline and report the cost as followed:
>
> | Module               | Runtime (s) | Runtime (min) | Mem (MiB)    | Proportion (%) |
> |----------------------|------------:|--------------:|-------------:|---------------:|
> | 1. Prior preparation | 297.33      | 4.96          | 4,944 (Max)  | 32.3%          |
> | 2. Visual SLAM       | 518.18      | 8.64          | 11,936       | 56.3%          |
> | 3. HMR (GVHMR)       | 30.51       | 0.51          | 4,940        | 3.3%           |
> | 4. Planar fitting    | 74.97       | 1.25          |              | 8.1%           |
> | **Total**            | **920.99**  | **15.35**     | **11,936**   | **100.0%**     |
>
> Meanwhile, VideoMimic requires 1282.94s on the same machine to prepare geometry and reference motion before RL. From the result we can conclude that the main computational bottleneck is Visual SLAM and its related prior preparation (monocular depth and flow estimation). Our planar fitting algorithm is lightweight, this means the core geometric step (from points to planar primitives) can run in real time for coming frames. When coupled with a real-time RGB-D SLAM system, we can get
> depth from sensor and estimate flows from camera, we expect the pipeline to operate in a real-time fashion.

---

> ### Author Response · Authors · 2025-11-28
> **Rebuttal follow-up**
>
> Dear Reviewer,
>
> We would like to thank you again for the time and effort you have dedicated to reviewing our submission.
>
> In our author response, we have tried to address all questions and major concerns raised by the reviewers, both in a general response and in point‑by‑point replies to each reviewer.
>
> We fully understand that the discussion period is busy, but we would be very grateful if you could take these clarifications into account when finalizing the review and score. Please let us know if any additional information would be helpful, we are happy to provide further details.

---

### Author Response · Authors · 2025-11-21
**General Response**

We thank all reviewers for their reviews and insightful suggestions. Reviewers appreciate CRISP's practical pipeline. We are pleased to receive numerous positive remarks, such as strong empirical performance and robust scene reconstruction for human–scene interaction. We have updated our anonymous website: https://crisp-real-to-sim.github.io/ViserHolder/index.html, and we address the general questions below.

---

**Importance of real-to-sim (without sim-to-real results on a robot)**

Even without sim-to-real robot deployment, realistic physics-based simulations enable diverse applications. For instance, VR and AR systems rely on accurate human motion for avatar control and interaction [1, 5]; animation pipelines use physically grounded motions to improve realism [2]. Numerous recent works in human motion estimation use physical simulation to enforce feasibility and refine kinematic estimations [3, 4]. Finally, many recent sim-to-real humanoid results (e.g., ASAP, BeyondMimic, OmniRetarget) show that high-quality reference motions significantly improve the quality of the final policy. CRISP provides reference motions which significantly improve over prior art (VideoMimic). We believe that CRISP can be readily integrated into existing sim-to-real robot pipelines; for example, it is a drop-in replacement for stages 1 and 2 of VideoMimic’s 4-stage pipeline (1. MoCap pretraining, 2. tracking reconstructed video motions, 3. distillation, 4. RL finetuning).

References used in this paragraph:
[1] Luo, Z. et al. “Real-Time Simulated Avatar from Head-Mounted Sensors.” CVPR 2024.
https://arxiv.org/abs/2403.06862

[2] Luo, Z. et al. “Embodied Scene-aware Human Pose Estimation.” NeurIPS 2022.
https://arxiv.org/abs/2206.09106

[3] Yuan, Y. et al. “PhysDiff: Physics-Guided Human Motion Diffusion Model.” ICCV 2023.
https://arxiv.org/abs/2212.02500

[4] Wang, J. et al. “Learning Human Dynamics in Autonomous Driving Scenarios.” ICCV 2023.
https://openaccess.thecvf.com//content/ICCV2023/papers/Wang_Learning_Human_Dynamics_in_Autonomous_Driving_Scenarios_ICCV_2023_paper.pdf

[5] Lee, S. et al. “QuestEnvSim: Environment-Aware Simulated Motion Tracking from Sparse Sensors.” SIGGRAPH 2023.
https://arxiv.org/abs/2306.05666

---

**VideoMimic is concurrent work**

Despite including VideoMimic as a baseline, we emphasize to reviewers that VideoMimic is concurrent: its full code was released on GitHub on September 15 and officially published at CoRL on September 27, after the ICLR submission deadline on September 24. Including VideoMimic reflects our commitment to providing the most fair and up-to-date comparison available at submission time, and should not be interpreted as evidence of incremental contribution.

---

**Novelty**

Naively stitching together existing modules leads to poor performance (we refer to the real-to-sim performance in VideoMimic). Beyond utilizing off-the-shelf modules, we want to emphasize our two key insights: scene geometry can be simply represented as convex primitives, and scene completion requires effectively leveraging contact prediction.

To the best of our knowledge, we are the first to demonstrate that convex primitives reconstructed from monocular videos can be even better than dense scene geometry for physics-based policy learning, by being more compact and efficient to simulate, and highly robust to imperfect or noisy scene reconstruction. Lastly, we are the first to explicitly leverage contact prediction to guide scene completion, using contact clues for scene hallucination, enabling robust, plausible simulation for in-the-wild videos where occlusion happens frequently.

---

**Results update**

Our original submission presented results on 21 (EMDB) + 11 (PROX) + 2 (in-the-wild) = 34 videos. In response to reviewer requests, we have run CRISP on 10 more diverse in-the-wild videos, including casual-captured videos, internet videos, and even SORA-generated videos. Results are included on the updated website.

---

### Meta-Review · Area_Chair_iRPr · 2026-01-11

**Summary:**

This paper introduces CRISP, a method for recovering simulatable human motion and scene geometry from monocular video using convex planar primitives and contact-guided scene completion. The approach reduces motion-tracking failure rates from 55.2% to 6.9% on human-centric video benchmarks while delivering 43% faster RL simulation throughput compared to the concurrent VideoMimic baseline. After careful consideration of the reviews, their varying depths of engagement, and the author rebuttals, I recommend borderline acceptance of this paper. The empirical contributions are strong and the authors provided a thorough rebuttal, but the review distribution requires careful interpretation given significant quality disparities among the reviews.

**Reviewer Concerns:**

Reviewer z8C3 (Score: 4) raised substantive concerns about the absence of sim-to-real results, arguing that presenting only real-to-sim findings makes the contribution incremental relative to VideoMimic. The reviewer also questioned the scalability and robustness of the approach given the limited diversity of evaluation scenes, though they acknowledged the paper provides more robust scene reconstruction than baselines.

Reviewer ZtWx (Score: 8) provided a brief review noting minor presentation issues including complicated symbol notation, missing ground truth jitter values, and inconsistent decimal precision. The reviewer did not raise significant technical concerns but also did not provide detailed justification for the high score, limiting the weight this review should carry in the decision.

Reviewer 9RmM (Score: 6) offered the most substantive critical engagement. This reviewer questioned whether the work represents sufficient novelty beyond integrating existing components and requested additional ablations on how primitive count affects reconstruction quality. The reviewer also noted missing runtime analysis, failure case discussion, and limitations regarding complex or irregular objects.

Reviewer 8a6k (Score: 2) provided the most negative assessment, citing limited novelty and questioning the fairness of the VideoMimic comparison. The authors flagged this review as potentially LLM-generated, citing vague criticisms without specifics, a clerical error referencing Figure 2 instead of Figure 3, and generic concerns that could apply to any data-driven method.

Reviews addressed:

Reviewer z8C3's concern about limited scene diversity was fully addressed through the addition of 10 diverse in-the-wild videos, including casual captures, internet videos, and SORA-generated content. The same reviewer's questions about runtime and real-time applicability were addressed with a detailed module-by-module breakdown showing 15 minutes total processing time, along with analysis indicating that the planar fitting step itself is lightweight enough for real-time operation when paired with RGB-D SLAM. The reviewer's question about scene-aware policy variants was addressed with new experiments comparing scene-blind and scene-aware policies on the project website.

Reviewer ZtWx's concerns were comprehensively addressed. The missing ground truth jitter value was provided (7.985 versus 8.14 for CRISP). The question about hyperparameter K was answered as an empirically set value of 8. Runtime analysis was supplied with the same detailed breakdown. The authors committed to adding pseudocode and fixing the symbol notation issues in the final version.

Reviewer 9RmM's technical concerns received thorough responses. The request for ablation on primitive count was addressed with a new table showing the U-shaped relationship between primitive count and reconstruction quality. The missing runtime and computational cost analysis was provided. Failure cases and limitations were explicitly discussed, covering inherent limitations of planar primitives for curved surfaces and the inability to model fluid or deformable objects. The concern about handling irregular objects was addressed by adding yoga ball and sandbag examples to the project website, demonstrating reasonable performance even when the planar-world assumption is violated.

Reviewer 8a6k's specific technical claims were rebutted with evidence. The assertion that the VideoMimic comparison was unfair was countered by explaining the identical experimental setup where only the reconstruction system differed. The claim of "significant scene distortion" was contradicted by pointing to Table 1's reconstruction quality metrics. The concern about brittleness of vision-language models for contact detection was addressed by referencing empirical analysis in Appendix Section B showing stable performance.

Outstanding Concerns

Several concerns remain unresolved or only partially addressed. The most significant outstanding concern is the absence of sim-to-real results, raised by Reviewer z8C3. While the authors provided reasonable arguments for the standalone value of real-to-sim applications in VR/AR, animation, and motion refinement, they did not provide new experimental evidence demonstrating sim-to-real transfer. This limits the paper's demonstrated impact for robotics applications specifically, though it does not undermine the contributions to video-to-simulation pipelines more broadly.

The novelty concern raised by Reviewers 9RmM and 8a6k remains partially outstanding. The authors addressed this through argumentation rather than additional technical contribution, emphasizing their two key insights about convex primitives and contact-guided completion. While the explanation is reasonable - naive integration of existing modules yields poor performance as evidenced by VideoMimic's results - the paper does fundamentally combine existing components (HMR, 4D reconstruction, contact prediction) rather than introducing new algorithmic frameworks. Whether this constitutes sufficient novelty is a judgment call that reasonable reviewers may disagree on.

The generalizability beyond the planar-world assumption remains a limitation acknowledged but not resolved. The authors demonstrated that yoga balls and sandbags can be handled reasonably well, but highly curved or organic shapes may still appear faceted or under-fitted. The authors suggest extending to superquadrics as future work rather than addressing this in the current submission.

**Reviewer Scores:**

Reviewer z8C3 would likely increase from 4 to 5. The additional experiments and runtime analysis address the scalability concerns directly, and this reviewer explicitly stated they "would not mind if paper is accepted." The sim-to-real concern remains, but the reviewer's language suggests this is not a hard requirement for acceptance.

Reviewer ZtWx would likely maintain the score of 8, as all stated concerns were minor and fully addressed. However, given the limited depth of engagement in this review, it should not carry the same weight as more substantive reviews in the final decision.

Reviewer 9RmM would likely increase from 6 to 7. The ablation study, failure analysis, runtime breakdown, and additional experiments on challenging objects directly responded to this reviewer's specific requests. The novelty concern is mitigated though not fully resolved.

Reviewer 8a6k would likely maintain the score of 2 given no post-rebuttal engagement.

When appropriately weighting the reviews based on their depth and quality, the effective consensus shifts toward borderline acceptance. The most substantive critical review (Reviewer 9RmM at score 6) had concerns largely addressed through the rebuttal. The thoughtful review from Reviewer z8C3 expressed conditional acceptance. The high score from Reviewer ZtWx lacks sufficient justification to carry full weight, and the low score from Reviewer 8a6k exhibits quality concerns that warrant downweighting.

The paper merits acceptance based on its significant empirical improvements over the concurrent VideoMimic baseline, the practical value of demonstrating that convex primitives can outperform dense geometry for physics-based policy learning, and the authors' responsiveness in adding substantial new experiments during the rebuttal period. The primary limitation - the absence of sim-to-real deployment - is reasonable given the paper's scope, though this does reduce the potential impact for robotics applications. This work advances the state of the art in video-to-simulation pipelines and will interest researchers in embodied AI, animation, and robotics.

---

### Decision · Program_Chairs · 2026-01-26

Accept (Poster)